# Analysis of *YUC* and *TAA/TAR* Gene Families in Tomato

Sida Meng [1,2,3,4], Hengzuo Xiang [1,2,3,4], Xiaoru Yang [1,2,3,4], Yunzhu Ye [1,2,3,4], Yuying Ma [1,2,3,4], Leilei Han [1,2,3,4], Tao Xu [1,2,3,4], Yufeng Liu [1,2,3,4], Feng Wang [1,2,3,4], Mingfang Qi [1,2,3,4,*] and Tianlai Li [1,2,3,4,*]

[1] College of Horticulture, Shenyang Agricultural University, Shenyang 110866, China; mengsida@syau.edu.cn (S.M.); xianghengzuo8308@163.com (H.X.)

[2] Modern Protected Horticulture Engineering & Technology Center, Shenyang Agricultural University, Shenyang 110866, China

[3] National & Local Joint Engineering Research Center of Northern Horticultural Facilities Design & Application Technology (Liaoning), Shenyang 110866, China

[4] Key Laboratory of Protected Horticulture, Shenyang Agricultural University, Ministry of Education, Shenyang 110866, China

[*] Correspondence: qimingfang@syau.edu.cn (M.Q.); ltl@syau.edu.cn (T.L.)

**Abstract:** Auxin is a vital phytohormone, but its synthesis pathway is poorly understood. This study used bioinformatic analysis to identify and analyze the gene family members that encode tomato auxin biosynthesis. The *FZY* gene family members encoding flavin-containing monooxygenases were retrieved from the tomato genome database. DNAMAN analysis revealed nine genes within the landmark domain WL(I/V)VATGENAE, between the FAD and NADPH domains. Phylogenetic analysis showed that the *FZY* gene family in tomato is closely related to the *YUC* gene family in *Arabidopsis thaliana*. A qRT-PCR showed that *SlFZY2*, *SlFZY3*, *SlFZY4-1*, and *SlFZY5* were highly expressed in tomato flower organs. The analysis of promoter cis-acting elements revealed light-responsive elements in the promoters of all nine members in tomato, indicating their sensitivity to light signals. Furthermore, the promoters of *SlFZY4-2*, *SlFZY5*, and *SlFZY7* contain low-temperature-responsive elements. This study demonstrated that *SlTAA5* expression was 2.22 times that of *SlTAA3* in the roots, and *SlTAA3* expression in the pistils was 83.58 times that in the stamens during the tomato flowering stage. Therefore, various members of the tomato *FZY* gene family are involved in regulating the development of tomato floral organs and are responsive to abiotic stresses, such as low temperature and weak light.

**Keywords:** tomato; auxin; *YUC* gene family; *TAA/TAR* gene family; bioinformatics

## 1. Introduction

Auxin participates in nearly every developmental process of plants, and its concentration gradient plays a key role in plant growth and development [1,2]. The site of auxin maxima around the apical meristem indicates the initiation site of leaf primordia [3]. Although the auxin response maxima are closely related to organ initiation and growth, the auxin minima play a crucial role in the formation of both axillary bud meristems and the separation layers of the *Arabidopsis thaliana* valve margin [4]. Auxin is synthesized locally and helps to optimize plant growth [5]. For example, after the auxin biosynthetic gene in the aboveground parts of rice was knocked out, the auxin in rice was significantly reduced, which resulted in increased rice tillers and a decreased seed setting rate [6–8]. The *A. thaliana yuc2yuc6* double mutant cannot develop functionally mature pollen, but its other developmental processes are the same as those in the wild-type plants [9]. The endoplasmic reticulum (ER) membrane localization for tryptophan aminotransferase, present in *Arabidopsis thaliana* (TAA)/YUC proteins involved in auxin biosynthesis, has appeared in the early evolution of bryophytes. ER membrane-anchored YUC proteins previously existed mainly in roots [10]. The SCF[TIR1/AFBs]-mediated signaling pathway participated in the feedback regulation of the YUC-mediated auxin biosynthesis pathway in *Arabidopsis*

*thaliana* [11]. It is obvious that *YUC* genes are necessary for developmental processes, from embryogenesis to seedling development to flower development. Overexpression of some *YUC* genes leads to similar phenotypes, suggesting that *YUC* may have overlapping functions; for example, the *YUC3*, *YUC5*, *YUC7*, *YUC8*, and *YUC9* genes were expressed in roots, and the inactivation of these five *YUC* genes (*yucQ*) led to the development of short and geotropic roots [12]. In addition to this, the auxin reporter DR5-GUS expression of *YUC1*, *YUC2*, *YUC4*, and *YUC6* were expressed in leaf primordia [13]; however, *YUC1* and *YUC4* show distinct and overlapping expression patterns, and even *yuc1 yuc4* double mutants do not exhibit any obvious defects in embryogenesis or the formation of leaves [8]. In *Arabidopsis thaliana*, the clavata1 (CLV1) receptor and histone acetyltransferase general control non-repressible 5 (GCN5) inhibited the expression *YUC4* by acetylating histone H3 [14]. SUPERMAN (SUP) interacted with polycomb repressive complex 2 (PRC2) and fine-tuned local auxin signaling by negatively regulating the expression of the auxin biosynthesis gene *YUC1/4* [15]. *YUC*-mediated auxin biosynthesis was also necessary for endosperm development in maize [16]. Microspore development in the *Arabidopsis thaliana yuc2yuc6* double mutant stalled before the first asymmetric mitotic division (PMI) of the pollen; therefore, the *yuc2yuc6* mutant could not produce viable pollen [17].

Plant endogenous auxins can be synthesized through the tryptophan (Trp)-dependent and Trp-independent pathways. However, studies on the Trp-independent pathway are limited, and this pathway is still poorly understood. Thus far, only one complete Trp-dependent pathway, the TAA/YUC pathway, has been established in plants [18–20], and it is highly conserved throughout the plant kingdom. The indole-3-acetic acid (IAA) balance in plants is coordinated through many processes. Firstly, the reactions catalyzed by TAA are reversible. Considering the high level of indole-3-pyruvate (IPA) [21], the aminotransferase VAS1 catalyzes IPA to Trp. During this process, methionine (Met) is catalyzed to $\alpha$-keto-$\gamma$-methylbutyric acid, which participates in ethylene biosynthesis. Secondly, when the IPA level is considerably elevated, another transaminase VAS1 converts IPA back to Trp to coordinate the biosynthesis of auxin and ethylene. Finally, IAA can be irreversibly oxidized and inactivated by dioxygenase for auxin oxidation. The biosynthesis and degradation of local auxin together maintain a steady state of endogenous auxin in plants to ensure their optimal growth [22]. Light and temperature are key factors for plant development. The DR5::GUS staining of apple seedlings after low-temperature treatment exhibited significantly decreased auxin in the roots [23]. Nevertheless, there are also reports of upregulating auxin with low temperature. For example, the IAA content in the flower spikes was significantly increased after wheat was treated at 4 °C for 21 days [16]. The effects of low temperature on auxin biosynthesis might be different for different species and organs [8,24]. The basic helix-loop-helix transcription factor, PHYTOCHROME-INTERACTING FACTOR 4 (PIF4) is a key regulator of plant thermomorphogenesis [25]. PIFs interact with phytochrome. The low ratio of red to far red light (R:FR) led to increased auxin levels in cotyledons [26].

Tomato (*Solanum lycopersicum*) is one of the most widely cultivated heat-loving vegetables. Fruit formation and development require a series of strict mechanisms, which is a complex and orderly process of biochemical and molecular changes. Auxin, as a plant hormone, plays an important role in regulating fruit growth and development. As a key gene family in the auxin biosynthesis pathway, the *YUCCA(YUC)* family has been widely studied in *A. thaliana*; however, reports on related studies in tomato are few. Therefore, this study applied bioinformatics to analyze and identify the key gene family *YUC/FZY* in the auxin biosynthetic pathway in tomato, and subcellular localization and cis-acting element prediction analysis were performed. Moreover, real-time quantitative PCR (qRT-PCR) was used to analyze the expression of *FZY* genes in various parts of tomato, and to clarify the members of the tomato *FZY* gene family that are involved in regulating the development of tomato floral organs and the response to abiotic stresses, such as low temperature and weak light stress.

## 2. Materials and Methods

### 2.1. Experimental Materials

The tomato variety tested in this experiment was 'Alisa Craig' (AC). The seeds were sown in the plug tray of an energy-saving solar greenhouse at the Facility Vegetable Research Station of Shenyang Agricultural University. The nighttime temperature was set to 15 °C, while the daytime temperature was 25 °C, with a photoperiod of 12/12 h. The humidity was 65%, and the illuminance was 400 $\mu mol\cdot m^{-2}\cdot s^{-1}$.

### 2.2. Experimental Methods

2.2.1. Identification and Naming of Tomato YUC and TAA Family Members

The amino acid sequences of the YUCCA and TAA family members were queried in the *A. thaliana* genome database (http://www.arabidopsis.org/, accessed on 25 April 2023); additionally, BLAST was conducted in the tomato genome database (https://solgenomics.net/, accessed on 25 April 2023) to obtain all tomato proteins with similar sequences to *A. thaliana* YUCCA and TAA family members. DNAMAN was used to compare the retrieved amino acid sequences, and the proteins containing the three functional domains of the YUCCA family were taken as the target family members for further verification. The phylogenetic relationship between the flavin-containing monooxygenase (FMO) and TAA families of tomato and the two families of *A. thaliana* was analyzed using the Maximum Likelihood method to construct a tree with MEGA5, and the resulting members were named according to their relationship distance.

2.2.2. Sequence Analysis of Tomato YUC and TAA Genes

The Gene Structure Display Server (GSDS, http://gsds.cbi.pku.edu.cn/, accessed on 25 April 2023) was adopted to analyze the *YUC* and *TAA* gene structures in tomato and *A. thaliana*. The MEME Suite (http://meme-suite.org/, accessed on 25 April 2023) was used to analyze the conservative motifs of these two families, and the TB tools toolkit was used to analyze the sequence conservation of the three functional domains of the YUC family. CELLO v.2.5: subCELlular LOcalization predictor (http://cello.life.nctu.edu.tw/, accessed on 25 April 2023) was adopted to predict the subcellular localization of tomato *YUC* family members. Ensembl Plants (http://plants.ensembl.org/index.html, accessed on 25 April 2023) was used to find the promoter sequences of the tomato *YUC* family, and the promoter database PlantCARE (http://bioinformatics.psb.ugent.be/webtools/plantcare/html/, accessed on 25 April 2023) was searched to analyze the promoter-binding elements for genes encoding the FMO family in tomato.

2.2.3. Extraction and Purification of Total RNA from Various Tissues of Tomato

The roots, stems, leaves, flower buds, fully open flowers, stamens, pistils, and young fruits of AC tomato were sampled under normal growth conditions from 60-day-old seedlings, and the pedicel abscission zones from newly opened flowers were cut into small segments (about 3 mm) using a sharp blade at anthesis. Three biological replicates for each sample were used. An RNA extraction kit from CWbiotech (Beijing, China) was used to extract total RNA from each sample, in preparation for later analysis of tomato *FZY* gene expression in various tissues. The total RNA concentrations were determined using a microplate reader, and their integrities were checked by agarose gel electrophoresis.

2.2.4. Preparation of cDNA from Various Tissues of Tomato

A reverse transcription kit from TAKARA (Kusatsu, Japan) was used to reverse transcribe the extracted total RNA. The reaction system included 4 $\mu L$ of RT master mix (5×) and 16 $\mu L$ of RNA + ddH$_2$O. After the reverse transcription program was completed, the cDNA templates were stored in a −20 °C freezer.

2.2.5. Analysis of Gene Expression in Various Tissues of Tomato

A real-time fluorescence qPCR kit from TAKARA was used to analyze gene expression. The reaction system included 10 μL of TB green advantage premix (2×); 2 μL of forward primer, 2 μL of reverse primer, 2 μL of cDNA, and 4 μL of ddH$_2$O. The relative gene expression was calculated from $2^{-\Delta\Delta Ct}$ values. A constitutively expressed *actin* gene (NCBI: NM_001330119.1) was used as a reference gene to normalize the cDNA. Each experiment was performed independently, two times, with at least three biological samples, and recorded with CFX96 real-time system (Bio-Rad, Hercules, CA, USA).

## 3. Results

### 3.1. Searching for and Naming of YUC/FZY Genes in Tomato

A total of 22 genes possibly homologous to the FMO-encoding *YUC* gene family in *A. thaliana* were found by BLAST in the tomato genome database. Further amino acid sequence alignment of these 22 genes with DNAMAN revealed nine genes containing the three functional domains: FAD (GAGPSGLA), NADPH (GCGNSGM), and WL(I/V)VATGENAE. Considering that the latter is the landmark domain of the FMO family (Figure 1), these nine proteins in tomato may belong to the FMO family. It follows that we identified *FZY* gene family members in tomato.

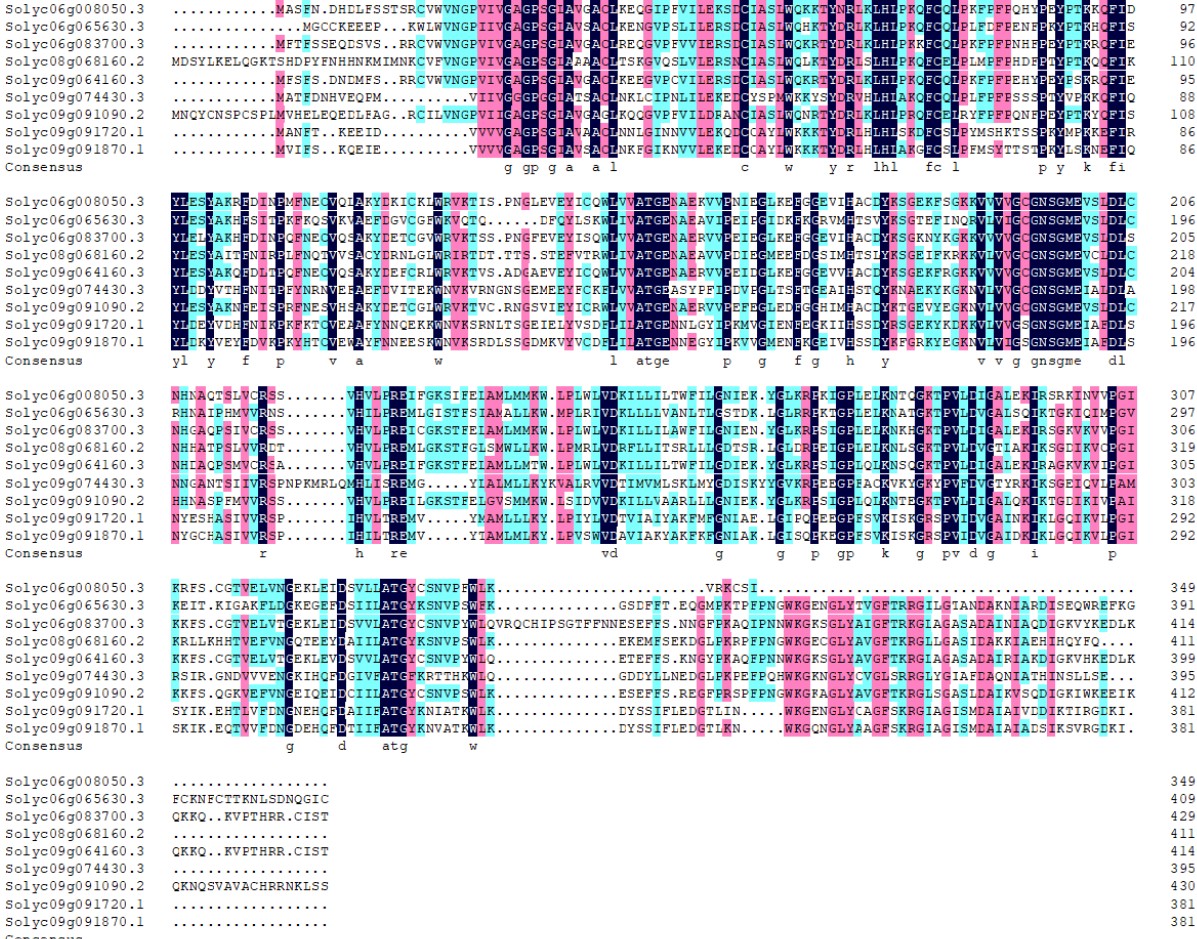

**Figure 1.** Amino acid sequence alignment of the nine target FZY proteins in tomato. Colors represent the homology of the sequences, and the darker the color, the higher the similarity.

### 3.2. Characteristic Analysis of YUC/FZY Genes in Tomato

The phylogenetic analysis of the proteins encoded by these nine genes in tomato and by the *YUC* family in *A. thaliana* showed a close phylogenetic relationship between the *FZY*

family of tomato and the *YUC* family of *A. thaliana* (Figure 2A). The tomato *FZY* family was named according to the phylogenetic results and previous research reports. The *FZY* family in tomato is closely related to the *YUC1* family present in *A. thaliana*. *Solyc08g068160*, which is closely related to *AtYUC2*, was named *SlFZY2*. *Solyc09g091090*, which is closely related to *AtYUC3* and *AtYUC7*, was named *SlFZY3*. *Solyc09g064160* was named *SlFZY4-1*, *Solyc06g008050* was named *SlFZY4-2*, and *Solyc06g083700* was named *SlFZY5*, and these genes are evolutionarily close to *AtYUC5*, *AtYUC8*, and *AtYUC9*. *Solyc09g074430* was named *SlFZY6*, *Solyc09g091720* was named *SlFZY7*, and *Solyc09g091870Y* was named *SlFZY8*; these genes are evolutionarily close to *AtYUC10* and *AtYUC11*. *SlFZY4-2*, *SlFZY7*, and *SlFZY8*, reported in this study, were newly discovered genes that have not been reported previously. We could also see that the nine YUC proteins in tomato could be grouped into four different groups (Figure 2A). *SlFZY3*, *SlFZY4-1*, *SlFZY4-2*, and *SlFZY5* comprised the first group. The three other groups are *SlFZY1*, *SlFZY2*, and *SlFZY6–SlFZY8*.

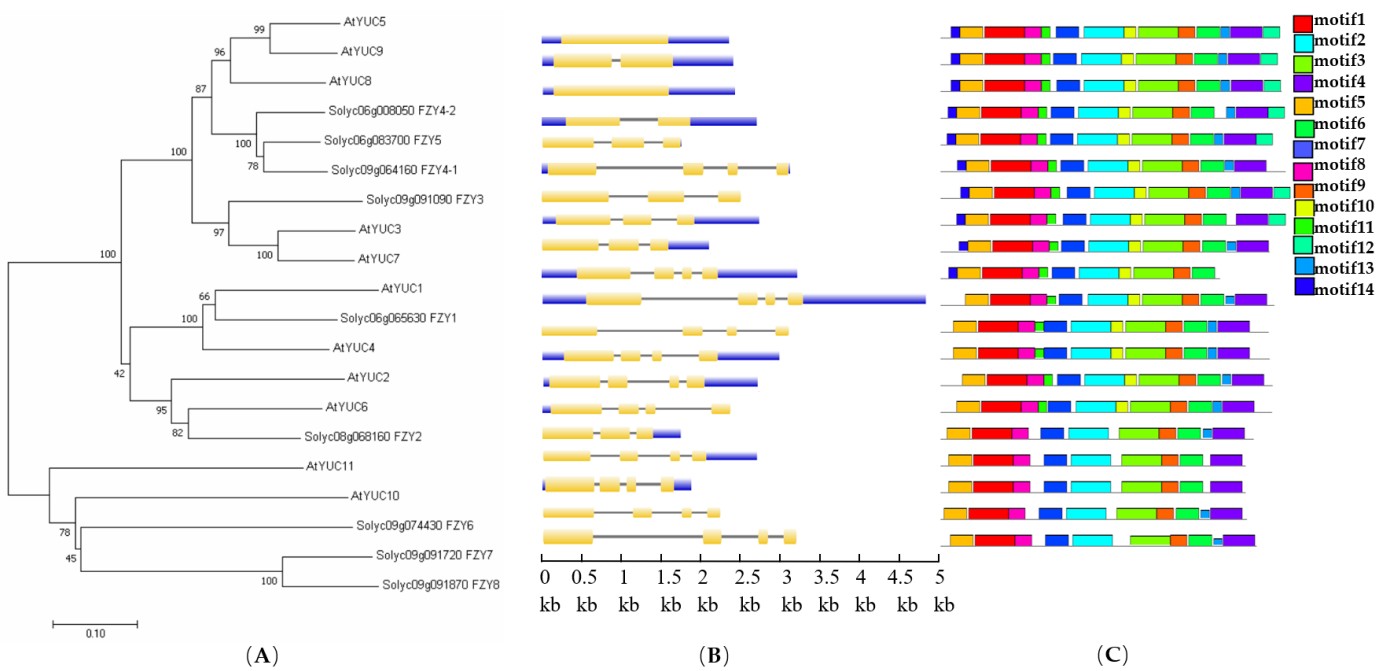

**Figure 2.** (**A**) Phylogenetic tree, (**B**) gene structure analysis, and (**C**) motifs of the tomato *FZY* gene family.

The online GSDS 2.0 was used to analyze the *FMO* gene family structure of tomato and *A. thaliana* (Figure 2B). The results showed that *SlFZY1*, *SlFZY2*, *SlFZY4-1*, *SlFZY6*, *SlFZY7*, *SlFZY8*, *AtYUC1*, *AtYUC2*, *AtYUC4*, *AtYUC6*, and *AtYUC10* each had four exons and three introns that do not encode proteins; *SlFZY3*, *SlFZY5*, *AtYUC3*, *AtYUC7*, and *AtYUC11* each had three exons and two introns; and *SlFZY4-2* and *AtYUC9* each had two exons and one intron, while *AtYUC5* had only one exon. As such, the auxin biosynthesis gene family members, which demonstrate a close homogeneous relationship between tomato and *A. thaliana*, have similar gene structures.

A MEME analysis of the protein structure of the YUC family demonstrated that motif19 is a shared motif (Figure 2C). *SlFZY6*, *SlFZY7*, *SlFZY8*, *AtYUC10*, and *AtYUC11* do not contain motif10 or motif11, a few genes do not contain motif12, and more than half do not contain motif13 or motif14.

The sequence logo (Figure 3) shows that the three functional domains of FAD, NADPH, and WL(I/V)VATGENAE are located in motif5, motif7, and motif2, respectively. FAD and NADPH are highly conservative, and WL(I/V) VATGENAE is conservative.

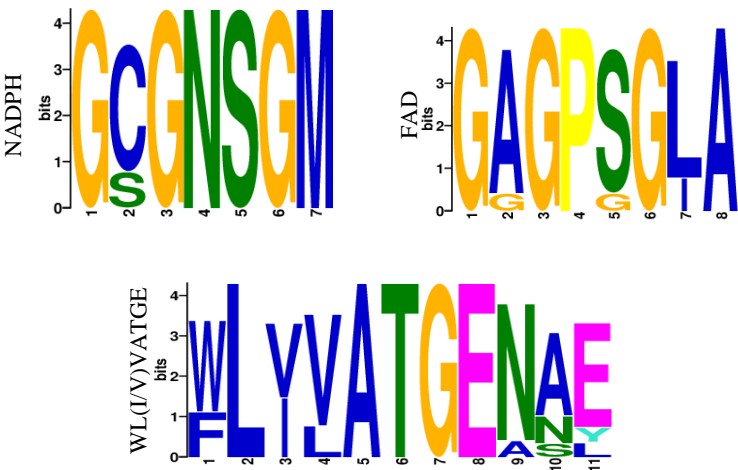

**Figure 3.** Amino acid sequence logo of the three functional domains.

The subcellular localization prediction of the tomato *FZY* gene family suggested that only *SlFZY2* might be present in the chloroplast, *SlFZY5* might be present in the cytoplasm or chloroplast, *SlFZY7* might be present in the cytoplasm or plasma membrane, and the six other proteins might be present in the cytoplasm (Table 1). Previous studies have shown that the TAA/YUC pathway completes the biosynthesis of auxin in the cytoplasm, which is consistent with our results.

**Table 1.** Subcellular localization prediction of tomato *SlFZY* gene family members.

| Gene Family | Gene Name | Serial Number | Subcellular Location |
|---|---|---|---|
| *YUCCA* | *SlFZY1* | *Solyc06g065630* | Cytoplasmic |
| | *SlFZY2* | *Solyc08g068160* | Mitochondrial |
| | *SlFZY3* | *Solyc09g091090* | Cytoplasmic |
| | *SlFZY4-1* | *Solyc09g064160* | Cytoplasmic |
| | *SlFZY4-2* | *Solyc06g008050* | Cytoplasmic |
| | *SlFZY5* | *Solyc06g083700* | Cytoplasmic/Mitochondrial |
| | *SlFZY6* | *Solyc09g074430* | Cytoplasmic |
| | *SlFZY7* | *Solyc09g091720* | Cytoplasmic/Plasma membrane |
| | *SlFZY8* | *Solyc09g091870* | Cytoplasmic |

Analysis of promoter cis-acting elements showed that only the two genes *SlFZY2* and *SlFZY8* do not contain CAAT-box. In addition (Table 2), *SlFZY8* does not contain TATA-box, indicating that it may lack transcriptional activity. The seven other genes contain these two cis-acting elements. As the core promoter element, the TATA-box is one of the binding sites of RNA polymerase. RNA polymerase binds to this site and initiates transcription. CAAT-box regulates the frequency of transcription initiation. This gene family may be susceptible to the environment because the promoters of all members contain light-responsive elements, indicating their sensitivity to light signals. Both *SlFZY1* and *SlFZY3* contain cis-acting elements of TC-rich repeats (Table 2) that function in defense and stress response. *SlFZY2* might be regulated by the MYB family of transcription factors to respond to drought stress because of the MBS-binding elements on its promoter (Table 2). The promoters of *SlFZY4-2* and *SlFZY6* contain the MYB recognition element that binds to MYB and participates in light response (Table 2). The promoters of *SlFZY4-2*, *SlFZY5*, and SlFZY7 contain low-temperature-responsive (LTR) elements, suggesting that low temperature may directly or indirectly regulate *FZYs* (Table 2). Synergistic or antagonistic interactions exist between hormones during the regulation of plant growth and development. *SlFZY2*, *SlFZY4-2*, *SlFZY5*, *SlFZY7*, and *SlFZY8* contain abscisic acid-responsive elements. All except *SlFZY1* contain the salicylic acid-responsive element TCA-element. *SlFZY3*, *SlFZY4-2*, and *SlFZY5*

contain the auxin-responsive element AuxRR-core. *SlFZY2*, *SlFZY3*, *SlFZY4-2*, *SlFZY5*, *SlFZY7*, and *SlFZY8* contain the methyl jasmonate-responsive element CGTCA-motif.

**Table 2.** Prediction of cis-acting elements of the *SlFZY* promoter.

| Gene | Regulatory Components | Number | Sequence | Function |
|---|---|---|---|---|
| *SlFZY1* | ATC-motif | 1 | AGTAATCT | part of a conserved DNA module involved in light responsiveness |
| | Box 4 | 7 | ATTAAT | part of a conserved DNA module involved in light responsiveness |
| | CAAT-box | 33 | CAAT/CAAAT | common cis-acting element in promoter and enhancer regions |
| | G-box | 1 | CACGAC | cis-acting regulatory element involved in light responsiveness |
| | GA-motif | 1 | ATAGATAA | part of a light-responsive element |
| | GATA-motif | 1 | AAGGATAAGG | part of a light-responsive element |
| | GT1-motif | 3 | GGTTAA | light-responsive element |
| | LAMP-element | 1 | CTTTATCA | part of a light-responsive element |
| | TATA-box | 26 | TATA/TATAA/ TATAAGAA/TATAAATA/ TATATTTATATTT | core promoter element around -30 of transcription start |
| | TC-rich repeats | 1 | GTTTTCTTAC | cis-acting element involved in defense and stress responsiveness |
| *SlFZY2* | AAGAA-motif | 1 | GAAAGAA | cis-acting element involved in abscisic acid responsiveness |
| | ABRE | 1 | ACGTG | cis-acting element involved in abscisic acid responsiveness |
| | ARE | 1 | AAACCA | cis-acting regulatory element essential for anaerobic induction |
| | TCA-element | 1 | TCAGAAGAGG | cis-acting element involved in salicylic acid responsiveness |
| | TCCC-motif | 1 | TCTCCCT | part of a light-responsive element |
| | TCT-motif | 1 | TCTTAC | part of a light-responsive element |
| | TGACG-motif | 1 | TGACG | cis-acting regulatory element involved in MeJA responsiveness |
| | I-box | 1 | AGATAAGG | part of a light-responsive element |
| | MBS | 1 | CAACTG | MYB binding site involved in drought inducibility |
| | P-box | 1 | CCTTTTG | gibberellin-responsive element |
| | CAT-box | 1 | GCCACT | cis-acting regulatory element related to meristem expression |
| | CGTCA-motif | 1 | CGTCA | cis-acting regulatory element involved in MeJA responsiveness |
| | G-box | 1 | CACGTC | |
| | GA-motif | 1 | ATAGATAA | |
| | GT1-motif | 1 | GGTTAA | |
| | Box 4 | 3 | ATTAAT | |
| | TATA-box | 83 | TATATA/TATA/ATT ATA/ATATAT/TATA CA/TATAAAA | |
| *SlFZY3* | A-box | 1 | CCGTCC | cis-acting regulatory element |
| | O2-site | 1 | GTTGACGTGA | cis-acting regulatory element involved in zein metabolism |
| | AuxRR-core | 1 | GGTCCAT | cis-acting regulatory element involved in auxin responsiveness |
| | TATA-box | 42 | TATA/ATATAT/TATAA ATA/TATAAA/TAT AA/TATAAAT | |
| | Box 4 | 8 | ATTAAT | |
| | CAAT-box | 43 | CAAAT/CAAT | |
| | TC-rich repeats | 1 | GTTTTCTTAC | |
| | G-box | 1 | CACGAC | |
| | GATA-motif | 1 | AAGATAAGATT | |
| | GT1-motif | 3 | GGTTAA/GGTTAAT | |
| | TCA-element | 1 | TCAGAAGAGG | |
| | CGTCA-motif | 1 | CGTCA | |
| | TCCC-motif | 1 | TCTCCCT | |
| | TGACG-motif | 1 | TGACG | |
| *SlFZY4-1* | Box 4 | 5 | ATTAAT | |
| | CAAT-box | 45 | CCAAT/CAAT/ CAAAT | |
| | TATA-box | 65 | TATA/ATATAT/TATA AATA/T ATAAA/T ATAA/TATAAAT | |
| | ARE | 1 | AAACCA | |
| | TCA-element | 1 | CCATCTTTTT | |

**Table 2.** *Cont.*

| Gene | Regulatory Components | Number | Sequence | Function |
|---|---|---|---|---|
| | MRE | 1 | AACCTAA | MYB binding site involved in light responsiveness |
| | ATCT-motif | 1 | AATCTAATCC | part of a conserved DNA module involved in light responsiveness |
| | MSA-like | 1 | TCAAACGGT | cis-acting element involved in cell cycle regulation |
| | LTR | 1 | CCGAAA | cis-acting element involved in low-temperature responsiveness |
| *SlFZY4-2* | CAAT-box | 32 | CAAAT/CAAT | |
| | LAMP-element | 1 | CTTTATCA | |
| | Box 4 | 4 | ATTAAT | |
| | G-Box | 3 | CACGTT | |
| | CGTCA-motif | 2 | CGTCA | |
| | TCA-element | 1 | TCAGAAGAGG | |
| | TGACG-motif | 2 | TGACG | |
| | I-box | 1 | ATGATAAGGTC | |
| | ARE | 1 | AAACCA | |
| | AuxRR-core | 1 | GGTCCAT | |
| | GARE-motif | 4 | TCTGTTG | gibberellin-responsive element |
| | chs-CMA1a | 3 | TTACTTAA | part of a light-responsive element |
| | GCN4_motif | 1 | TGAGTCA | cis-regulatory element involved in endosperm expression |
| | GATA-motif | 2 | GATAGGG/GATAGG | |
| | G-box | 5 | GCCACGTGGA/CACGTG/ CCACGTAA/TACGTG | |
| | GT1-motif | 7 | GGTTAAT/GGTTAA | |
| *SlFZY5* | Box 4 | 5 | ATTAAT | |
| | TATA-box | 77 | TACAAAA/ATATAT/TATA/TATAA/ TATAAAA/TATAAA/TATATAA | |
| | CAAT-box | 48 | CAAT/CCAAT/CAACCAACTCC/CAAAT | |
| | TCA-element | 1 | CCATCTTTTT | |
| | TCT-motif | 1 | TCTTAC | |
| | ARE | 3 | AAACCA | |
| | I-box | 1 | GGATAAGGTG | |
| | TGACG-motif | 2 | TGACG | |
| | CGTCA-motif | 2 | CGTCA | |
| | ABRE | 3 | CACGTG | |
| | LTR | 1 | CCGAAA | |
| | AT-rich sequence | 1 | TAAAATACT | element for maximal elicitor-mediated activation |
| | AT-rich element | 1 | ATAGAAATCAA | binding site of AT-rich DNA-binding protein (ATBP-1) |
| | TATA-box | 53 | TATTTAAA/ATATAT/ TATA/ATTATA/TATAA/TAAAGATT/ TATATA/TATATTTATATTT/TATAAAT | |
| *SlFZY6* | Box 4 | 1 | ATTAAT | |
| | CAAT-box | 20 | CAAT/CAAAT | |
| | GT1-motif | 1 | GGTTAA | |
| | ARE | 3 | AAACCA | |
| | TCCC-motif | 1 | TCTCCCT | |
| | TCA-element | 1 | CCATCTTTTT | |
| | MRE | 1 | AACCTAA | |
| | chs-CMA1a | 2 | TTACTTAA | |
| | ACA-motif | 1 | AATTACAGCCATT | part of gapA in (gapA-CMA1) involved with light responsiveness |
| | 3-AF1 binding site | 1 | TAAGAGAGGAA | light-responsive element |
| | WUN-motif | 1 | AAATTTCCT | wound-responsive element |
| | Gap-box | 2 | CAAATGAA(A/G)A | part of a light-responsive element |
| | TATA-box | 33 | TATAA/TATA/ATATAT/ TATATA/ATATAA/TATACA/TATAAAA | |
| *SlFZY7* | G-Box | 1 | CACGTT | |
| | Box 4 | 3 | ATTAAT | |
| | CAAT-box | 40 | CAAT/CAAAT/CCAAT | |
| | TGACG-motif | 1 | TGACG | |
| | TCA-element | 1 | CCATCTTTTT | |
| | ABRE | 1 | ACGTG | |
| | CGTCA-motif | 1 | CGTCA | |
| | LTR | 1 | CCGAAA | |
| | ATCT-motif | 1 | AATCTAATCC | |
| | G-box | 4 | TACGTG/GCCACGTGGA | |
| | TGACG-motif | 2 | TGACG | |
| | CGTCA-motif | 2 | CGTCA | |
| *SlFZY8* | ABRE | 5 | ACGTG/GACACGT GGC/CACGTG/ | |
| | CAT-box | 1 | GCCACT | |
| | ARE | 2 | AAACCA | |
| | TCA-element | 1 | CCATCTTTTT | |

It follows that the *FZY* family of tomato is closely related to the *YUC* family of *Arabidopsis*. Subcellular localization prediction of the tomato *FZY* family revealed that eight

of its members may be in the cytoplasm and one in the chloroplast. Through the analysis of promoter cis-acting elements, it was found that promoters of all members of this gene family in tomato contain photoresponsive elements, indicating that they are sensitive to light signals. Only *SlFZY4-2*, *SlFZY5*, and *SlFZY7* have low-temperature response elements on their promoters.

### 3.3. Analysis of Expression Patterns of YUC/FZY Genes in Various Tissues of Tomato

Local biosynthesis of auxin regulates the growth and development of plants; the *YUC/FZY* gene is a key rate-limiting enzyme in auxin synthesis. Therefore, to clarify the role of the *YUC/FZY* gene in tomato growth and development, total RNA from various tissues of tomato was extracted and reverse transcribed. The expression of *SlFZY* genes in various tomato parts was analyzed by qRT-PCR (Figure 4). The results showed a spatiality in the expression pattern of the *SlFZY* gene family. *SlFZY3* was the most highly expressed in the underground parts of tomato. *SlFZY5* was the most highly expressed in the stems and in the leaves. In the fully open flowers, *SlFZY3* was the most highly expressed, followed by *SLFZY5*. In the flower buds, *SLFZY2* expression was the highest. *SLFZY3* and *SLFZY4-1* were highly expressed in stamens and pistils. *SlFZYs* were lowly expressed in the flower stalks. In fruits, *SLFZY4-2* was highly expressed, with high tissue specificity. In conclusion, the *YUC/FZY* gene is involved in the whole process of tomato growth and development.

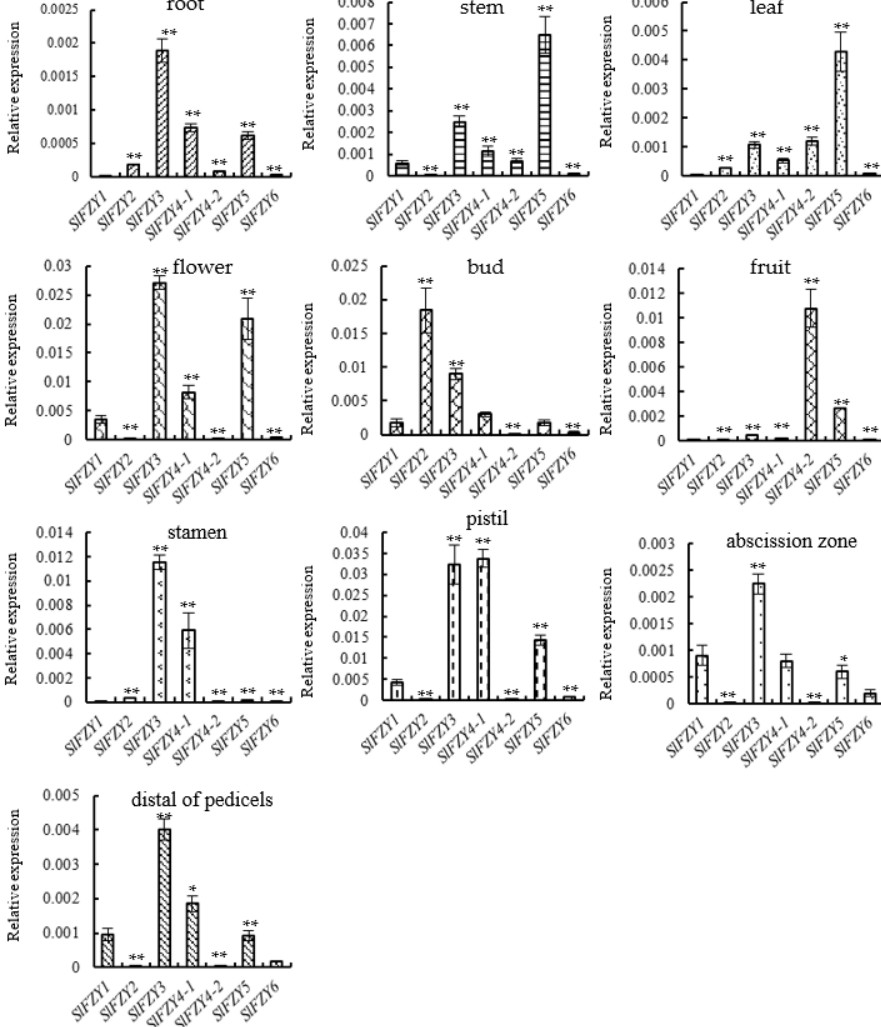

**Figure 4.** qRT-PCR analysis of *SlFZYs* relative expression levels normalized to *actin* in various tissues of tomato. Significant difference was determined by one-way ANOVA with Dunnett's test; * indicates significant difference ($p < 0.05$); ** indicates extremely significant difference ($p < 0.01$).

### 3.4. Identification and Structure Analysis of the Genes Encoding TAA Transaminase in Tomato

The role of the TAA aminotransferase family in plants is crucial in auxin biosynthesis; however, this family has not been reported in tomato (Figure 5). In the present study, a total of five possible members of this family was retrieved in the tomato genome database. Phylogenetic analysis showed that these five genes in tomato are closely related to the *TAA/TAR* genes in *A. thaliana*. Phylogenetic nomenclature was performed, and *Solyc01g017610* was named *SlTAA1*, *Solyc02g062190* was named *SlTAA2*, *Solyc03g112460* was named *SlTAA3*, *Solyc05g031600* was named *SlTAA4*, and *Solyc06g071640* was named *SlTAA5*. Gene structure analysis revealed that only *SlTAA2* contains four exons, and all other genes have five exons. The motif analysis results were similar to the gene structure analysis results. The structure of the SlTAA2 protein is remarkably different from those of other proteins, containing only seven motifs. It follows that we identified *TAA* gene family members in tomato.

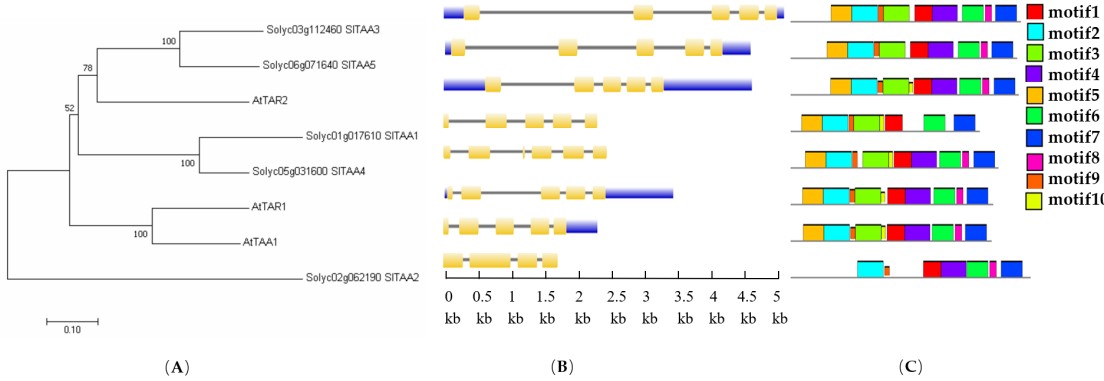

**Figure 5.** (**A**) Phylogenetic tree based on alignment of protein amino acid sequences; (**B**) Structural analysis of the five target genes in tomato and three *TAA/TAR* genes in *Arabidopsis thaliana*; and (**C**) Motif analysis of the five target genes in tomato and three *TAA/TAR* genes in *Arabidopsis thaliana*.

### 3.5. Expression Analysis of TAA Genes in Various Tissues of Tomato

The catalysis of tryptophan to indole pyruvate by TAA transaminase in plants is the first step of the TAA/YUC pathway. NCBI was used to design primers for the TAA family. Only the primers of *SlTAA3* and *SlTAA5* were specific, while the three other genes were far from the *A. thaliana* TAA transaminase family phylogenetically. Hence, in this study, the expressions of *SlTAA3* and *SlTAA5* from various tissues of tomato were quantitatively analyzed by qRT-PCR (Figure 6). The results showed that the expression of *SlTAA3* was lower than that of *SlTAA5* in the roots, as that of *SlTAA5* was 2.22 times that of *SlTAA3*. In other tomato tissues, the expression of *SlTAA3* was higher than that of *SlTAA5* and reached its peak in the flower buds. During the tomato flowering stage, the expression of *SlTAA3* in the pistils was 83.58 times that in the stamens. In conclusion, the *TAA* gene, especially *SlTAA3*, is involved in the whole process of tomato growth and development.

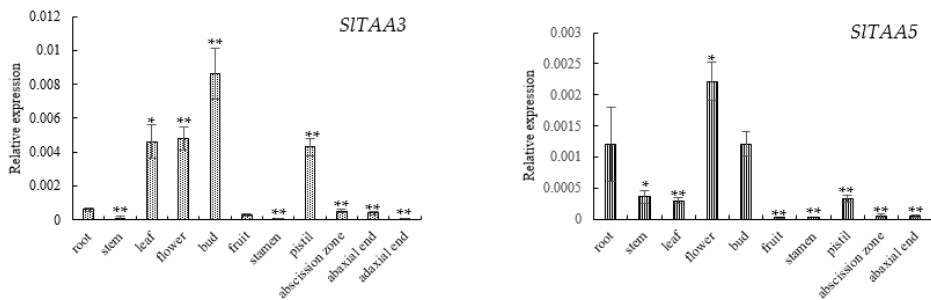

**Figure 6.** qRT-PCR analysis of *SlTAA3* and *SlTAA5* relative expression levels normalized to *actin* in various tissues of tomato. Significant difference was determined by one-way ANOVA with Dunnett's test; * indicates significant difference ($p < 0.05$); ** indicates extremely significant difference ($p < 0.01$).

## 4. Discussion

Auxin and its locality are accompanied by the whole process of plant growth and development; the transportation and signal transduction of auxin in tomato have been studied, but its biosynthesis has not been studied. In this study, a bioinformatics analysis was conducted to find key genes encoding rate-limiting enzymes in the auxin synthesis pathway in tomato. Bioinformatic analysis was used to successfully identify nine *YUCCAs* (*FZYs*) in the tomato genome, all of which contain the WL(I/V)VATGENAE domain, located between FAD and NADPH domains. Considering that this domain is the landmark domain of the FMO family, the nine genes possibly encode FMOs and participate in the auxin biosynthesis pathway in tomato. The resulting phylogenetic tree showed that the *FZY* gene family in tomato is closely related to the *YUC* gene family, with 11 members in in *A. thaliana*. It is additionally similar that tomato *FZY* proteins can be also grouped into four groups, as *YUC* ones can in *A. thaliana* [10]. *YUC* is the rate-limiting enzyme in the auxin biosynthesis pathway in plants. Thus, finding the genes encoding this enzyme in tomato is crucial for every aspect of tomato research. The subcellular localization prediction of the tomato *FZY* gene family revealed that eight of its members might be present in the cytoplasm and one in the chloroplast. Auxin biosynthesis occurs in the cytoplasm [27], suggesting that these nine genes are involved in the biosynthesis of tomato auxin. Promoter cis-acting element analysis exhibited light-responsive elements in the promoters of all members of this gene family in tomato, suggesting their sensitivity to light signals. Previous studies have shown that, in *A. thaliana*, PIF4 could regulate *YUC4* expression in response to light signals [28]. Our results are consistent with this conclusion. The promoters of *SlFZY4-2*, *SlFZY5*, and *SlFZY7* contain LTR elements. It is speculated that low temperature might regulate tomato development by affecting the expression of these three genes. LTR elements were not found on the promoters of the other genes, but they may regulate *SlFZY* through other transcription factors in tomato. Through bioinformatic analysis, this research further identified five TAA/TRA aminotransferase family genes in the tomato genome. The qRT-PCR results showed that *SlTAA3* was highly expressed in tomato, suggesting its crucial role in the growth and development of tomato. Phylogenetically, *SlTAA3* is closely related to *AtTAR* in *A. thaliana*, and their gene structure and motif modules share high similarities. The TAA family produces IPA and the YUC family functions in the conversion of IPA to IAA in *A. thaliana* through a quantification process of IPA [27]. Two gene families encoding key enzymes in the TAA/YUC auxin biosynthesis pathway were found in tomato, and they provided a new direction for further exploring the regulatory mechanism of auxin in tomato development. Plant auxins are synthesized locally [29]. Analyzing the expression pattern of *SlFZYs* in various tissues of tomato is an important part of their functional study. It is well known that over-expression of *YUC* genes increases auxin production in *A. thaliana*, while the loss of function of a single *YUC* gene does not affect plant growth [30]. Through qRT-PCR analysis, it was found that *SlFZY2*, *SlFZY3*, *SlFZY4-1*, and *SlFZY5* are highly expressed in tomato flower organs, similar to *YUC1* and *YUC4* in *A. thaliana*. These four genes are speculated to regulate the development of tomato floral organs. The highest expression of *SlFZY3* was in the underground parts of tomato, that of *SlFZY5* was highest in the stems and leaves, and that of *SlFZY4-2* was highest in young fruits. It is well known that the root tip is an important component for the synthesis of auxin, and the concentration of auxin can affect the change in the root tip structure. It has been confirmed that the mutation of *YUC3*, *YUC 5*, *YUC 7*, *YUC 8* and *YUC 9*, which are highly expressed in the root system of *Arabidopsis thaliana*, will interfere with root growth. Therefore, it can be inferred that the *SlFZY3* in the tomato root system may be involved in the formation of root structure [10,31]. However, unlike in the study of *Arabidopsis*, the function of the *FZY* gene in tomato remains unclear. However, bioinformatic analysis could not fully prove that the *FZY* gene family in tomato is responsible for encoding FMOs, and further confirmation via the *iaam* gene complementation test is needed. The *YUC* gene family shows serious functional redundancy in *A. thaliana*. Thus, it is to be determined by the virus-induced gene silencing tool whether members of the tomato *FZY* gene family show

distinct and overlapping expression patterns. Further experiments are needed to verify the gene function.

## 5. Conclusions

Auxin is involved in nearly all processes of plant growth and development, and its locality is essential. In this study, the *FZY* gene family members for tomato auxin biosynthesis were identified using bioinformatic methods. The expressions of such genes in various tissues of tomato were analyzed. The spatiality of their expression pattern was demonstrated, and their expression was the highest in the flower organs of tomato. The analysis of promoter cis-acting elements revealed light-responsive elements in the promoters of all found gene family members in tomato, indicating their sensitivity to light signals. The promoters of *SlFZY4-2*, *SlFZY5*, and *SlFZY7* contain LTR elements. In addition, the expression of *SlTAA5* was 2.22 times that of *SlTAA3* in the roots. The expression of *SlTAA3* in the pistils was 83.58 times that in the stamens during the tomato flowering stage. Therefore, our study demonstrated that various members of the tomato *FZY* gene family are involved in regulating the development of tomato floral organs and the response to abiotic stresses, such as low temperature and weak light.

**Author Contributions:** Data curation, S.M. and H.X.; investigation, X.Y., Y.Y. and Y.M.; methodology, T.X., Y.L. and F.W.; project administration, M.Q. and T.L.; Validation, X.Y. and L.H.; writing—original draft, S.M. and H.X.; writing—review and editing, S.M., H.X., X.Y., M.Q. and T.L. All authors have read and agreed to the published version of the manuscript.

**Funding:** This research was funded by the National Natural Science Foundation of China, grant numbers 32102460, 31972397, and 32172554; as well as the 2021 Scientific Research Funding Project of Liaoning Provincial Department of Education, grant number LJKZ0639.

**Data Availability Statement:** The authors will supply the relevant data in response to reasonable requests.

**Acknowledgments:** The authors are grateful to the National & Local Joint Engineering Research Center of Northern Horticultural Facilities Design & Application Technology (Liaoning), the Modern Protected Horticulture Engineering & Technology Center (Shenyang Agricultural University), and the Key Laboratory of Protected Horticulture (Shenyang Agricultural University) Institute for supporting this project.

**Conflicts of Interest:** The authors declare no conflict of interest.

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
