# Peer review of "Analysis of YUC and TAA/TAR Gene Families in Tomato"

_horticulturae, doi:10.3390/horticulturae9060665_

Round 1

Reviewer 1 Report

Analysis of YUC and TAA/TAR Gene Family in Tomato

Sida Meng, Hengzuo Xiang, Xiaoru Yang, Yunzhu Ye, Yuying Ma, Leilei Han, Yufeng Liu, Feng Wang, Mingfang Qi, Tianlai Li, Tao Xu   

Comments

Introduction

The authors indicate in the introduction that a "correlation between DR5 expression and YUC expression pattern was also seen in the seedling development. YUC1, YUC2, YUC4, and YUC6 were expressed in leaf primordia" and cite Zhang [1]. In the reviewer's opinion, this is an incorrect interpretation of what was said in the article, which is a review of the literature. First, Zhang cites two papers in this regard [2,3], and second, the DR5 reporter is expressed regardless of whether there is an expression of YUC or any other gene. DR5 is a reporter of the auxin response. In other words, it is expressed if there is auxin linked to the TGTCTC sequence, which allows the expression of the reporters attached to the said promoter, as reported by Liao et al., I quote below "A widespread reporter of auxin response, the synthetic DR5 promoter, consists of 7–9 TGTCTC AuxRE repeats and marks sites of transcriptional auxin response by activating reporters such as β-glucuronidase, fluorescent proteins [4].” This section of the introduction must be rewritten.

Another misquoted reference is that of Eklund et al. [5]. They refer to the role of the STYLISH1 gene family in increasing YUCCA family transcript levels, and “their results suggest that STY1, and most likely other SHI/STY members, are DNA binding transcriptional activators that target genes encoding proteins mediating auxin biosynthesis[5].” They do not refer to the clavata1 receptor or histone acetyltransferase.

The introduction will need to be rewritten to place the correct citations.

The format of the references is not correct. The year is attached to the name of the magazine in all cases. The DOI is not complete either. It must be written in full, e.g., http://doi.org/10.1105/tpc.108.064816

400 μmol·m-2·s-1. Must be 400 μmol·m-2·s-1.

ddH2O. Must be ddH2O

The discussion should include comparisons with published similar works [6-10]. For example, how do you compare the number of genes found? Are the genes expressed in tomatoes in the different organs of the plant also expressed in the same organ in other plants, or is their expression exclusive to tomatoes? Is there any relationship between the auxin content and the expression of the YUCCA genes? This last question is very transcendent since, in various models, it has been seen that this relationship exists.

The language of the manuscript requires a significant revision. This reviewer has edited the abstract as an example.

Abstract. Auxin is a vital phytohormone, but its synthesis pathway is poorly understood. This study used bioinformatic analysis to identify and analyze the gene family members encoding tomato auxin biosynthesis. The FZY gene family members encoding flavin-containing monooxygenases were retrieved from the tomato genome database. DNAMAN analysis revealed nine genes with the landmark domain WL(I/V)VATGENAE between FAD and NADPH domains. Phylogenetic analysis showed that the FZY gene family in tomato is closely related to the YUC gene family in Arabidopsis thaliana. qRT-PCR showed that SlFZY2, SlFZY3, SlFZY4-1, and SlFZY5 were highly expressed in tomato flower organs. The analysis of promoter cis-acting elements revealed light-responsive elements in the promoters of all nine members in the tomato, indicating their sensitivity to light signals. Furthermore, the promoters of SlFZY4-2, SlFZY5, and SlFZY7 contain low-temperature responsive elements. This study demonstrated that SlTAA5 expression was 2.22 times that of SlTAA3 in the roots, and SlTAA3 expression in the pistils was 83.58 times that in the stamens during the tomato flowering stage. Therefore, various members of the tomato FZY gene family are involved in regulating the development of tomato floral organs and are responsive to abiotic stresses, such as low temperature and weak light.

References

        1.    Zhao, Y.  The role of local biosynthesis of auxin and cytokinin in plant development. Curr. Opin. Plant Biol. 2008, 11, 18-22, http://dx.doi.org/10.1016/j.pbi.2007.10.008.

        2.    Cheng, Y., Dai, X., Zhao, Y.  Auxin biosynthesis by the YUCCA flavin monooxygenases controls the formation of floral organs and vascular tissues in Arabidopsis. Gene. Dev. 2006, 20, 1790-1799, http://doi.org/10.1101/gad.1415106.

        3.    Cheng, Y., Dai, X., Zhao, Y.  Auxin synthesized by the YUCCA flavin monooxygenases is essential for embryogenesis and leaf formation in Arabidopsis. Plant Cell 2007, 19, 2430-2439, http://doi.org/10.1105/tpc.107.053009.

        4.    Liao, C.Y., Smet, W., Brunoud, G., Yoshida, S., Vernoux, T., Weijers, D.  Reporters for sensitive and quantitative measurement of auxin response. Nat. Meth. 2015, 12, 207-210, https://doi.org/10.1038/nmeth.3279.

        5.    Eklund, D.M., StaIdal, V., Vasecchi, I., Cierlik, I., Eriksson, C., Hiratsu, K., Ohme-Takagi, M., Sundstrom, J.F., Thelander, M., Ezcurra, I. et al.  The Arabidopsis thaliana STYLISH1 protein acts as a transcriptional activator regulating auxin biosynthesis. Plant Cell 2010, 22, 349-363, http://doi.org/10.1105/tpc.108.064816.

        6.    Xu, D., Miao, J., Yumoto, E., Yokota, T., Asahina, M., Watahiki, M.  YUCCA9-mediated auxin biosynthesis and polar auxin transport synergistically regulate regeneration of root systems following root cutting. Plant Cell Physiol. 2017, 58, 1710-1723, https://doi.org/10.1093/pcp/pcx107.

        7.    Bernardi, J., Battaglia, R., Bagnaresi, P., Lucini, L., Marocco, A.  Transcriptomic and metabolomic analysis of ZmYUC1 mutant reveals the role of auxin during early endosperm formation in maize. Plant Sci. 2019, 281, 133-145, https://doi.org/10.1016/j.plantsci.2019.01.027.

        8.    Han, X.L., Zhao, F.Y., Wang, Z.L., Che, X., Cui, G.C.  The role of OsYUCCA2 in auxin synthesis and promotion of rice growth and development. Russ. J. Plant Physiol. 2020, 67, 1018-1027, https://doi.org/10.1134/S1021443720060072.

        9.    Luo, W., Xiao, N., Wu, F., Mo, B., Kong, W., Yu, Y.  Genome-wide identification and characterization of YUCCA gene family in Mikania micrantha. Int. J. Mol. Sci. 2022, 23, 13037, https://doi.org/10.3390/ijms232113037.

      10.    Uc-Chuc, M.Á., Ku-González, Á., Jiménez-Ramírez, I.A., Loyola-Vargas, V.M.  Identification, analysis and modeling of the YUCCA protein family in genome-wide of Coffea canephora. Proteins 2022, 90, 1005-1024, https://doi.org/10.1002/prot.26293.

The English must be revised.

Author Response

Response to Reviewer 1 Comments

Point 1: The authors indicate in the introduction that a "correlation between DR5 expression and YUC expression pattern was also seen in the seedling development. YUC1, YUC2, YUC4, and YUC6 were expressed in leaf primordia" and cite Zhang [1]. In the reviewer's opinion, this is an incorrect interpretation of what was said in the article, which is a review of the literature. First, Zhang cites two papers in this regard [2,3], and second, the DR5 reporter is expressed regardless of whether there is an expression of YUC or any other gene. DR5 is a reporter of the auxin response. In other words, it is expressed if there is auxin linked to the TGTCTC sequence, which allows the expression of the reporters attached to the said promoter, as reported by Liao et al., I quote below "A widespread reporter of auxin response, the synthetic DR5 promoter, consists of 7–9 TGTCTC AuxRE repeats and marks sites of transcriptional auxin response by activating reporters such as β-glucuronidase, fluorescent proteins [4].” This section of the introduction must be rewritten.

Response 1: The introduction has been revised to correct the reviewer’s suggestion that there is no necessary link between the DR5 and YUC genes. The expression is changed to “the auxin reporter DR5-GUS expression of YUC1, YUC2, YUC4, and YUC6 were expressed in leaf primordia [13], however, YUC1 and YUC4 show distinct and overlapping expression patterns and even yuc1 yuc4 double mutants do not exhibit any obvious defects in embryogenesis and the formation of leaves”, and we have added relevant content in the introduction.

Point 2: Another misquoted reference is that of Eklund et al. [5]. They refer to the role of the STYLISH1 gene family in increasing YUCCA family transcript levels, and “their results suggest that STY1, and most likely other SHI/STY members, are DNA binding transcriptional activators that target genes encoding proteins mediating auxin biosynthesis [5].” They do not refer to the clavata1 receptor or histone acetyltransferase.

Response 2: The reference citation is wrong, and we have made a modification in the new manuscript.

Point 3: The format of the references is not correct. The year is attached to the name of the magazine in all cases. The DOI is not complete either. It must be written in full, e.g., http://doi.org/10.1105/tpc.108.064816

Response 3: We have revised the format of the references in the new manuscript and added the complete DOI.

Point 4: 400 μmol·m-2·s-1. Must be 400 μmol·m-2·s-1. ddH2O. Must be ddH2O

Response 4: We have corrected these errors in the new manuscript.

Point 5: The discussion should include comparisons with published similar works [6-10]. For example, how do you compare the number of genes found? Are the genes expressed in tomatoes in the different organs of the plant also expressed in the same organ in other plants, or is their expression exclusive to tomatoes? Is there any relationship between the auxin content and the expression of the YUCCA genes? This last question is very transcendent since, in various models, it has been seen that this relationship exists.

Response 5: We have added to the discussion the number of FZY genes found in tomatoes to compare the number of FZY genes found in Arabidopsis. But because the function of the FZY gene in tomatoes is unclear, it is not possible to compare the function of the YUC gene in Arabidopsis and other species.

Reviewer 2 Report

1) Introductions need more information and why YUCCA(YUC) are important in tomato?

2) Method:

1) The roots, stems, leaves, flower buds, fully open flowers, stamens, pistils, and young fruits: specify plant developmental stage while collecting these tissues and how many replications were used to collect the samples?

2) How about abscission zone sample was collected, can not found any detail in the method section?

3)  What is an abaxial end sample in tissue specific gene expression?

4) How did you perform q-rt-pcr, what was your house keeping gene, and machine used? How did you obtain relative expressions?

 3) Result:

1) Phylogenetic tree was not making sense to me again, for Instance,

a) How did you decide to give Solyc08g068160 as FZY2 instead FZY6? Both AtYUC6 and AtYUC2 were closely related to Solyc08g068160

b)  How AtYUC10/11 linked to FZY6-8, why you have not mentioned it FZY 10/11? 

2) What is the rationale for performing tissue-specific gene expression analysis with these sets of genes? Please provide an explanation at the start of the results section. And what do you want to draw attention to after gene expression?

3) Another point of concern: if there is a 1 CT difference, the fold change will be at least 2-fold, but your relative expression data was in 0.00xxx, which does not make sense to me; please check and you can see the expression of selected genes using insilico using gene investigator or another platform.

4) Since the expression of SlTAA3 in the pistils was 83.58 times that in the stamens during the tomato flowering stage, why not compare both qrt-pcr (SlTAA3 and SlTAA5 across all tissues) and then show 84-fold? It is only 0.00xxx relative expression in the current graphical illustration.

5)  What would be your concluding remark of SlTAA3 and SlTAA5 across all the tissues gene expression, in one line at the end of each result description.

6) Can you support your gene expression with previously available information or insilico analysis either in Arabidopsis or tomato?

7) Statistics are not used in the Qrt-pcr section.

8) In each result section, write one line of closing remarks.

1)      Scan the entire manuscript for English, grammatical, and typographical errors.

Author Response

Response to Reviewer 2 Comments

Point 1: Introductions need more information and why YUCCA(YUC) are important in tomato?

Response 1: In the introduction, we added the importance of studying the FZY gene in tomatoes. Because FZY gene is a homologous gene of Arabidopsis YUC gene, and YUC gene is closely related to the content of auxin, which affects the whole process of plants from seedling to flowering to fruit development. It is particularly important to study the role of YUC gene in tomatoes.

Point 2: Method:

1) The roots, stems, leaves, flower buds, fully open flowers, stamens, pistils, and young fruits: specify plant developmental stage while collecting these tissues and how many replications were used to collect the samples?

2) How about abscission zone sample was collected, can not found any detail in the method section?

3)  What is an abaxial end sample in tissue specific gene expression?

4) How did you perform q-rt-pcr, what was your house keeping gene, and machine used? How did you obtain relative expressions?

Response 2: Method:

1) We have added sampling time and number of replicates to the method.

2) We have added sampling methods for abscission zone samples to the new manuscript.

3) The abaxial end sample statement is incorrect and has been corrected.

4) We have added the qRT-PCR calculation method, house keeping gene, and the instrument company and model used in the new manuscript.

Point 3: Result:

1) Phylogenetic tree was not making sense to me again, for Instance,

  1. a) How did you decide to give Solyc08g068160 as FZY2 instead FZY6? Both AtYUC6 and AtYUC2 were closely related to Solyc08g068160?
  2. b)  How AtYUC10/11 linked to FZY6-8, why you have not mentioned it FZY10/11

2) What is the rationale for performing tissue-specific gene expression analysis with these sets of genes? Please provide an explanation at the start of the results section. And what do you want to draw attention to after gene expression?

3) Another point of concern: if there is a 1 CT difference, the fold change will be at least 2-fold, but your relative expression data was in 0.00xxx, which does not make sense to me; please check and you can see the expression of selected genes using insilico using gene investigator or another platform.

4) Since the expression of SlTAA3 in the pistils was 83.58 times that in the stamens during the tomato flowering stage, why not compare both qrt-pcr (SlTAA3 and SlTAA5 across all tissues) and then show 84-fold? It is only 0.00xxx relative expression in the current graphical illustration.

5)  What would be your concluding remark of SlTAA3 and SlTAA5 across all the tissues gene expression, in one line at the end of each result description.

6) Can you support your gene expression with previously available information or insilico analysis either in Arabidopsis or tomato?

7) Statistics are not used in the qRT-PCR section.

8) In each result section, write one line of closing remarks.

Response 3: Result:

1) The phylogenetic tree analysis of the FZY gene was carried out to refer to Arabidopsis and preliminarily clarify the function of the tomato YUC gene we screened. The initial idea of our numbering was to sort genes with conserved domains from smallest to largest by gene number, refer to the gene structure, and then number them according to their evolutionary relationship with Arabidopsis. The designation of Solyc08g068160 instead of FZY6 is due to the fact that Solyc08g068160 and YUC2, YUC6 evolved on the same branch, so the smaller numbered YUC2 was chosen. FZY6-8 is numbered according to the previous numbering to FZY5, so the numbers 6-8 are continued, and 10 and 11 are not selected.

2) We selected the identified FZY gene for tissue-specific analysis. The reason for not selecting FZY8 was that FZY8 may not have transcriptional activity as mentioned earlier. We have added the purpose of the study before this part of the results, and clearly stated that gene expression analysis is performed to study the role of the FZY gene in tomato growth and development.

3) We described the expression of genes relative to actin, that is, relative expression normalized to actin, and we have checked that the data multiples are more than 2 times.

4) At the tomato flowering stage, the expression of SlTAA3 in the pistils is 83.58 times that of the stamens. We represented the expression relative to actin in the graph, so there was only a relative expression of 0.00xxx in the current graphical illustration. We focused on the expression of auxin synthesis genes in flower organs, so only the fold difference between the two in the pistils was calculated.

5) The conclusion of SlTAA3 and SlTAA5 in all tissue gene expression is to screen key TAA genes in flower organs, so as to bind to YUC genes and identify key pathways affecting auxin synthesis in flower organs.

6) The data in http://tomexpress.toulouse.inra.fr/ can support our expression data.

7) We have added data significance analysis in the new manuscript.

8) We have written one line of closing remarks in each result section.

Point 4: Scan the entire manuscript for English, grammatical, and typographical errors.

Response 4: We have scanned the entire manuscript for English, grammatical, and typographical errors.

Reviewer 3 Report

Manuscript “Analysis of YUC and TAA/TAR Gene Family in Tomato” is dealing with a key genes in the auxin biosynthesis pathway. Presented study applied bioinformatics to analyze and identify the key gene family YUC/FZY in the auxin biosynthetic pathway in tomato, subcellular localization and cis-acting element prediction analysis were performed. The study demonstrated that various members of the tomato FZY gene family are involved in regulating the development of tomato floral organs and response to abiotic stresses, such as low temperature and weak light. The results of this study represent valuable base for further auxin gene function validation in tomato

All errors which can be corrected are highlighted in text. There are one extra blank page after Figure 3

Author Response

Point: All errors which can be corrected are highlighted in text. There are one extra blank page after Figure 3.

Response: We have corrected all the highlighted errors in the new manuscript and removed the blank pages.

Reviewer 4 Report

In this manuscript, the authors aim to identify and analyze the genes encoding Flavin-containing monooxygenase (FMO) family proteins (YUC/FZY) and Tryptophan aminotransferase (TAA) in tomato (Solanum lycopersicum), as these genes are known to play pivotal roles in auxin biosynthesis. To achieve this objective, the authors employed a range of bioinformatics tools, including BLAST, DNAMAN, GSDS 2.0, MEME, and qRT-PCR, to identify and characterize these genes in tomato and explore their phylogenetic relationship with Arabidopsis thaliana.

The study successfully identified nine FMO genes and five TAA genes in tomato. Phylogenetic analysis revealed a close relationship between the FZY family in tomato and the YUC family in A. thaliana. Additionally, gene structure analysis demonstrated that members sharing a close phylogenetic relationship between tomato and A. thaliana possess similar gene structures. Subcellular localization predictions suggested the cytoplasm as the primary location for these proteins, corroborating findings from previous studies. The authors also examined the promoter cis-acting elements and expression patterns of YUC/FZY genes across various tomato tissues.

 Despite providing a comprehensive analysis of FZY and TAA genes in tomato, the paper presents several notable limitations:

 1.The figure legends lack sufficient detail, hindering a clear understanding of the presented data.

2.The phylogenetic analysis is limited in scope, only encompassing two species (A. thaliana and tomato). Expanding the analysis to include other species could strengthen the study's conclusions.

3.The manuscript's methodology section is lacking in detail, and in some instances, relies on outdated software. For example, MEGA5 was used for phylogenetic analysis, while the current version is MEGA11. The authors should clarify how alignments were performed, the methods employed to infer phylogeny, and how relative expression was calculated in qPCRs (normalizing gene, equipment used, etc.). Additionally, it is important to specify if biological replicates were used in the experiments and, if so, their number.

4.The authors only analyzed the expression patterns of two TAA genes (SlTAA3 and SlTAA5) due to primer specificity issues. Resolving these issues and analyzing the expression patterns of all five TAA genes could provide a more comprehensive understanding of their role in auxin biosynthesis.

5.The paper lacks functional validation of the identified FZY and TAA genes. Incorporating experimental approaches, such as gene knockout or overexpression, would offer greater insight into their functional roles in auxin biosynthesis.

6.The bioinformatic analysis alone cannot fully substantiate the claim that the FZY gene family in tomato is responsible for encoding FMOs. Further validation through the iaam gene complementation test is necessary.

7.The authors speculate that the tomato FZY gene family may exhibit functional redundancy, akin to the YUC gene family in A. thaliana. However, additional experiments are needed to confirm gene function.

8.It is recommended that the authors investigate interactions between the identified FZY and TAA proteins and other proteins involved in auxin biosynthesis to enhance understanding of their roles in the auxin biosynthesis pathway.

Author Response

Response to Reviewer 4 Comments

Point 1: The figure legends lack sufficient detail, hindering a clear understanding of the presented data.

Response 1: Some figure legends have been modified in the new manuscript.

Point 2: The phylogenetic analysis is limited in scope, only encompassing two species (A. thaliana and tomato). Expanding the analysis to include other species could strengthen the study's conclusions.

Response 2: We only selected Arabidopsis as a phylogenetic tree for comparison, in order to infer the function of the YUC gene in tomato with reference to the function of Arabidopsis, so we did not select other species.

Point 3: The manuscript's methodology section is lacking in detail, and in some instances, relies on outdated software. For example, MEGA5 was used for phylogenetic analysis, while the current version is MEGA11. The authors should clarify how alignments were performed, the methods employed to infer phylogeny, and how relative expression was calculated in qPCRs (normalizing gene, equipment used, etc.). Additionally, it is important to specify if biological replicates were used in the experiments and, if so, their number.

Response 3: The method of constructing the phylogenetic tree has been added. The software, methods and instruments involved in qPCRs have been supplemented.

Point 4: The authors only analyzed the expression patterns of two TAA genes (SlTAA3 and SlTAA5) due to primer specificity issues. Resolving these issues and analyzing the expression patterns of all five TAA genes could provide a more comprehensive understanding of their role in auxin biosynthesis.

Response 4: We performed the experiment after designing specific quantitative primers, but none of the primers for the remaining TAA genes peaked.

Point 5: The paper lacks functional validation of the identified FZY and TAA genes. Incorporating experimental approaches, such as gene knockout or overexpression, would offer greater insight into their functional roles in auxin biosynthesis.

Response 5: This study only used bioinformatics to preliminarily analyze the function of FZY gene in tomatoes and didn’t carry out functional verification. We will conduct more in-depth research on this issue in follow-up trials.

Point 6: The bioinformatic analysis alone cannot fully substantiate the claim that the FZY gene family in tomato is responsible for encoding FMOs. Further validation through the iaam gene complementation test is necessary.

Response 6: Thanks for your comments, and we will further supplement this part of the experiment to publish the paper.

Point 7: The authors speculate that the tomato FZY gene family may exhibit functional redundancy, akin to the YUC gene family in A. thaliana. However, additional experiments are needed to confirm gene function.

Response 7: Thanks for your suggestions, we will further supplement this part of the experiment.

Point 8: It is recommended that the authors investigate interactions between the identified FZY and TAA proteins and other proteins involved in auxin biosynthesis to enhance understanding of their roles in the auxin biosynthesis pathway.

Response 8: At present, this study can only achieve preliminary screening of key FZY and TAA genes. The relationship between the synthesis of auxin precursors by TAA gene and then the synthesis of auxin by the rate-limiting enzyme FZY cannot be completed at present. Thanks for your valuable comments.

Round 2

Reviewer 1 Report

The manuscript has improved substantially after review and can be accepted after correcting the journal names in the bibliography. Some names are abbreviated as PNAS (citation 22), and the full name must be entered; while other titles are spelled with mixed case (citation 30; Annual review of plant biology), the correct spelling must be Annual Review of Plant Biology).

Author Response

Thanks for your comments, we have revised the format of the references in the new manuscript.

Reviewer 2 Report

The manuscript made significant improvement in this updated version,  I encourage editor to accept it in its present form. 

Author Response

Thanks for your comments. We will strive for even greater improvements in the next revision process.

Reviewer 4 Report

The authors have not adequately addressed all the concerns raised in the previous round of review. Specifically, Figures 4 and 6 remain insufficiently detailed in their captions for proper interpretation. The stability of the actin gene in this study is still not justified with supporting references, and the identifier for this gene in tomato is not provided. A request was made for the inclusion of the equipment used for the qPCR assays, and the equipment cited (T100) is an endpoint thermocycler, not a real-time one.

The authors need to better justify the novelty of their findings, particularly in light of similar studies such as: Bioinformatics Analysis of Phylogeny and Transcription of TAA/YUC Auxin Biosynthetic Genes - PMC (nih.gov)

Author Response

Thanks for your comments. We have supplemented the questions you raised in the new manuscript.

  1. We further explain the figures in the captions of Figures 4 and Figure 6.
  2. We provided the identifier of the actin gene in tomatoes.
  3. The name of the equipment used for qPCR assays has been modified in the new manuscript.
  4. We have cited more references to better justify the novelty of the findings.

If you have any further questions about the content of the manuscript, please don’t hesitate to contact us at any time for discussion and revision.